# Families with neurodevelopmental diagnoses are not 'Hard to Reach': Findings from a feasibility trial comparing parenting programmes for parents of children with ADHD

Fiona Sim[1], Lindsay Dalgarno[1], Emma McIntosh[1], Caroline Haig[2], Patrycja Duklas[2], Alex McConnachie[2], Christopher Gillberg[1,3], Helen Minnis[1,3‡], Lucy Thompson[1,3,4‡*]

1 School of Health and Wellbeing, University of Glasgow, Glasgow, Scotland, 2 Robertson Centre for Biostatistics, School of Health and Wellbeing, University of Glasgow, Glasgow, Scotland, 3 Gillberg Neuropsychiatry Centre, University of Gothenburg, Gothenburg, Sweden, 4 Institute of Applied Health Sciences, University of Aberdeen, Inverness, Scotland

‡ Denotes joint senior authorship
* helen.minnis@glasgow.ac.uk

## Abstract

Attention-Deficit/Hyperactivity Disorder (ADHD) is defined by impairing levels of inattention and/or hyperactivity-impulsivity. It occurs in at least 5% of school age children and 2.5% of adults worldwide, and is associated with multiple negative outcomes throughout life. There is good evidence to support the efficacy of pharmacological treatment of individuals affected, and also of behavioural training for parents of children with ADHD, though the optimal focus and format to support change has yet to be established. This feasibility, parallel-group, randomised, controlled, pilot trial examined the feasibility of comparing two parenting programmes for families of school-aged children with ADHD. Parents of children aged 5–12 years with a clinical diagnosis of ADHD, referred to a regional integrated ADHD pathway, were randomly allocated (1:1 in permuted blocks of 4) either to a Parents InC or Incredible Years (IY) parenting group. The primary feasibility outcomes were recruitment and retention rates. The putative primary effectiveness outcome measure was Parenting Sense of Competence Scale (PSOC) and secondary outcome measures included the behavioural, physical, and emotional functioning of the child as well as health and ADHD symptoms of the parents at 12 (or 6 in final four participants) months post-randomisation. Process and economic evaluations were also included. We recruited 30/52 (58%) eligible participants (14 randomised to Parents InC, 16 to IY). Twenty-two participants (73%) provided follow-up quantitative data and 24 (80%) participated in qualitative interviews. PSOC scores were suggestive of greater improvement in Parents InC groups than IY groups. Power calculations suggest we will need to recruit 230 participants for a definitive RCT. Health economic analysis showed Parents InC had a lower per-group costs. Participant feedback on research procedures

**Data availability statement:** In line with ethical research practices and to uphold participant confidentiality, we have ensured that all personal identifiers have been removed or anonymised in our dataset. However, due to the sensitivity of the information and the potential risk of re-identification of children and parents through indirect identifiers contained in the small study dataset, we will not be able to publish the full dataset. We obtained informed consent from all participants for the use of their data in this study. However, consent for public data sharing was not sought, as this was not part of the original study protocol. Consequently, we cannot make the dataset publicly available. Data requests may be sent to Dr Debra Stuart (debra.stuart@glasgow.ac.uk).

**Funding:** This work was funded by a research grant from the Scottish Government Health Directorates' Chief Scientist Office (https://www.cso.scot.nhs.uk/), ref: HIPS/17/58. The funder had no direct involvement in the design, data collection, analysis, decision to publish, or preparation in the manuscript.

**Competing interests:** The authors have declared that no competing interests exist.

and methods was positive, and interviews and process evaluation provided a rich source of learning points to take forward into a future trial. **Trial registration** Clinical Trials, protocol registration system: NCT03832270.

## Introduction

Attention-Deficit/Hyperactivity Disorder (ADHD) is defined by impairing levels of inattention and/or hyperactivity-impulsivity [1] and demonstrates a high level of familial occurrence [2]. ADHD is associated with multiple negative outcomes throughout life [3–8]. Prevalence estimates suggest ADHD occurs in approximately 5% of school age children and 2.5% of adults worldwide [7]. In childhood it can manifest in difficulty organising tasks and sustaining attention during schoolwork or play, frequently losing belongings and an inability to stay seated to a degree inconsistent with age or developmental level [1], with impact on social relationships and educational attainment [7]. ADHD displays a high level of symptomatic overlap with other neurodevelopmental diagnoses including developmental coordination disorder, language disorder, autism and intellectual disability [9]. Externalising disorders such as oppositional defiant disorder and conduct disorder show a particularly high level of co-occurrence with ADHD in childhood [7] presenting challenges not only to the individual experiencing the symptoms, but also to those responsible for their care and education [7,10,11].

Many parents manage well when the child has ADHD, and there is evidence that maternal ADHD can enhance the care-giving relationship [12,13]. However, challenges in the care-giving relationship resulting from ADHD symptoms can express themselves in negative communication styles, increased permissiveness and over-reactivity and lower responsivity than in families of children without ADHD [14–18]. Parenting a child with ADHD can be stressful and has been associated with child maltreatment [2,19]. Current best practice recommendations for the treatment of ADHD in children over 5 years of age outline a three tiered approach of supporting parents through giving information on ADHD, including via ADHD-focused parenting groups, pharmacotherapy in children aged 6 years and older, and Cognitive Behavioural Therapy [20]. The efficacy of behavioural training for parents of children with ADHD symptoms is well established in the general population [21–24], though there is some indication that the presence of parental ADHD symptoms may mediate the efficacy of parent training for children with ADHD [25–27].

Factors which determine the efficacy of a parenting intervention, such as attendance, engagement and subsequent consistent application of learning, present real challenges to parents particularly when they themselves have ADHD symptoms: difficulty with organisation and with conforming to routine are likely to be fundamental [28]. Low recruitment and retention to parenting programmes have significant impacts on their effectiveness [29] and barriers such as parental psychopathology, stigma, childcare and programme/service barriers have been identified as key [30–33]. These barriers can appear almost insurmountable for families who, for personal (e.g., education, psychopathology) and situational (e.g., deprivation, homelessness)

circumstances, consistently struggle to access or engage in services [34]. If services are to be accessible and inclusive, then clinical research regarding effectiveness must ensure that marginalised families are able to participate.

Many of the best-evidenced programmes offered as usual care require a significant time commitment from parents and may not always fit their needs. Health organisations may offer more generic 'behaviour management' parenting programmes rather than an ADHD-specific intervention as a means to covering many bases with a single point of investment. This can lead to gaps in service provision, which can result in smaller, less well tested interventions being implemented to fit an apparent need. Such programmes may show good user acceptability and positive impact in a simple pre- post-intervention analysis. One such programme is Parents In Control, or Parents InC [35]. Developed over the past 15 years or so, Parents InC offers specific support around empowerment, information and behaviour management specific to ADHD, as well as understanding of the child's development context. Parents InC was developed in Scotland, has been used for a number of years already and has evaluated well [35,36], but with relatively small sample sizes, no long-term follow-up, no economic evaluation and, most crucially, no comparison to an alternative intervention or to a control group. Our pilot work suggests Parents InC may be a much more cost-effective way of supporting parents compared to existing programmes. We therefore now need to understand if Parents InC is: i) is at least as effective as the current best-evidenced alternative, Incredible Years, in impacting children's behaviour outcomes; ii) is cost-effective; and iii) offers something helpful and unique compared to other parenting programmes in terms of parenting self-competence and quality of life.

This article presents findings from a feasibility randomised controlled trial (RCT) comparing two parenting interventions for parents of children with an ADHD diagnosis. The interventions were: Parents In Control (Parents InC) [35], designed specifically for parents of children with an ADHD diagnosis with a particular focus on parental sense of self competence; and the Incredible Years (IY) [37], a group parenting support intervention aimed at strengthening parent-child interactions and attachment, reducing harsh discipline and fostering parents' ability to promote children's social, emotional, and academic development. IY is established as an effective intervention programme for parents of children with disruptive behaviour problems, with positive outcomes in child behaviour, parent self-efficacy, and parent wellbeing [21,38,39]. Parents InC is delivered over six sessions to parents of children with a diagnosis of ADHD and IY is delivered over fourteen sessions to parents of children with a range of behavioural concerns. Preliminary evaluation of the programme has been positive, with some evidence of improvement in parent-reported frequency and intensity of child problem behaviours in a pre-post study [40]. At the time of conducting this research, Parents InC is the first-line parenting programme for parents of a child with a diagnosis of ADHD in NHS Fife.

The aims of this study were to identify whether a) parents of children recently diagnosed with ADHD were willing to be randomised to either Parents InC or Incredible Years, and b) sufficient numbers of families could be recruited and retained such that a full-scale randomised controlled trial is likely to be feasible. The following secondary research questions were addressed, to inform future definitive trial design, and will be reported here:

- Are research procedures and efficacy measures feasible and acceptable to participating families?

- Do families participating in Parents InC achieve similar scores on the parenting sense of competence scale at 12 months post randomisation as those in the comparison arm (IY)?

- Do the two intervention arms significantly differ on any other measures?

- What is the mean cost per participant of Parents InC?

## Materials and methods

### Trial design

We conducted a feasibility parallel-group RCT pilot study, comparing two parenting programmes for families of school-aged children with ADHD. An equal allocation ratio of 1:1 was adopted for this trial in order to most accurately reflect what would happen in a future definitive RCT.

## Participants

Parents of 5–12-year-old children with a formal diagnosis of ADHD were eligible for inclusion. The children had had a standardised assessment from a paediatrician or a psychiatrist after referral to the Fife integrated ADHD pathway between January 2019 and January 2020.

Families attending other parenting group-based interventions were excluded, as were parents who had low proficiency in English (as this could compromise ability to complete research measures or participate in a group intervention). Participants already taking part in research on a parenting intervention were also ineligible to participate.

Target recruitment figures were based on known clinical throughput within the Fife integrated ADHD pathway and on known recruitment and retention rates from a similar intervention trial conducted in a more socioeconomically deprived population [41]. In order to obtain follow-up data on at least 40 participants (20 per arm), we expected to need to approach 100 families in total (assuming a 60% response-rate, i.e., n = 60 randomised, 30 per arm; and two thirds of participating families being available for follow-up).

## Procedure

All participants were recruited within the NHS Fife area between 1st January 2019 and 31st January 2020. Those eligible to participate, who were part of the Fife integrated ADHD pathway, were informed about the study through one of three channels: 1) during a routine appointment with referring paediatricians and psychiatrists; 2) an assistant clinical psychologist within the ADHD pathway telephoned families on the waiting list for Parents InC groups; or 3) An invitation to participate in Parents InC groups containing a tear-off 'Expression of Interest [research]' form was sent to the parents' home address by the clinical psychology team; eligible participants who returned this form were then contacted directly by the trial team. Referrers were provided with eligibility criteria to ensure appropriate referrals to the research trial. Eligibility was confirmed by the research team on receipt of the referral information.

A participant information sheet and consent form were then posted to the home of potential participants and followed up with a telephone call within one week of the information being posted to allow informed written consent to be obtained. During this call, participants were asked to return the freepost consent form and an appointment was made to collect baseline data. Participants were given the choice of their preferred location for data collection, and the options were as follows: 1) face-to-face interview at the participant's home; 2) face-to-face interview in a clinical assessment room at an NHS facility in Fife; or 3) remotely via telephone or video call. Baseline data collection took 60–90mins, after which participants were randomised using an Interactive Voice Response System at the clinical research facility. Participants were allocated 1:1 (in permuted blocks of 4 with no stratification or minimisation) to either Parents InC or IY, and referral information was emailed to the relevant intervention team by the clinical research facility staff. A teacher-report copy of the Strengths & Difficulties Questionnaire (SDQ) was left with the parent/carer at the baseline visit for the parent to pass on to the child's class teacher; these were returned to the trial office by the class teacher via freepost envelope.

Two separate 3-month iterations of recruitment/ intervention were planned to allow time for reflection and refinement for each iteration. A third round of intervention was planned for any remaining participants who were unable to take up participation in the first group they were offered. A subsequent decision was taken to include a fourth iteration in response to referral delays. Participants in the first three recruitment iterations received follow-up data collection 12 months post-randomisation. Those participants in the fourth iteration received theirs at 6 months post randomisation. In response to the nationally enforced social distancing measures brought into force on the 23rd March 2020 during the international COVID19 pandemic, the trial moved entirely to remote data collection and all follow-up data was collected via telephone or video call.

## Qualitative methods

Semi-structured interviews were conducted with consenting trial participants at baseline and 12 months post-randomisation, and interviews held with intervention practitioners during and after the intervention.

Qualitative methods were employed to explore the following areas:

- Acceptability of research procedures to participating families;

- Quality of life for families living with ADHD;

- Parental expectations and experiences of intervention;

- Acceptability of research procedures to participating practitioners.

## Measures

The *primary intervention efficacy outcome*, for a future definitive RCT is likely to be parents' sense of competence measured by the Parental Sense of Competence Scale (PSOC) [42]. The PSOC is a 17-item questionnaire showing good validity across several studies and widely used as a measure of parenting self-esteem [43,44], collected at baseline and at 12-months post-randomisation.

The following potential *secondary intervention outcome measures* were incorporated into a user-friendly questionnaire booklet (around 250 items) and collected at baseline and at follow-up interviews:

- Eyberg Child Behaviour Inventory (ECBI) [45]: measure of behaviour problems of children aged 2–16 to be completed by parents or carers;

- SDQ [46]: self-report measure of behaviour of children aged 2–17 to be completed by parents/carers/teachers/children aged 11yrs or older;

- General Health Questionnaire (GHQ) [47]: self-report measure of common psychiatric conditions to be completed by parents or carers;

- ADHD Symptom Rating Scale-version 1 (ASRSv1) [48]: symptom checklist consisting of 18 DSM-IV criteria to be completed by practitioners with regards parents;

- Parenting Daily Hassles Scale (PDHS) [49]: 18-item self-report measure of the frequency and intensity of 20 potential parenting 'hassles' to indicate parents' stress;

- Goal Based Outcomes (GBO) from the Child Outcomes Research Consortium (CORC) [50]:

- EQ-5D-5L [51]: a self-report measure of parent quality of life;

- PedsQL [52]: a self-report measure of child quality of life;

- Participant & child health service use; a checklist of inpatient & outpatient services accessed within the last 12 months.

## Analysis

A simple descriptive analysis of numbers of families recruited and retained relative to the eligible/ approached population was conducted. Between-group change in each measure after adjustment for baseline was analysed using analysis of covariance (ANCOVA). This allowed exploration of preliminary effects of the intervention and will help decide on the sample size needed for a future definitive RCT. Missing data were not imputed, and all available data was used in

all analyses. All analyses were governed by a pre-specified Statistical Analysis Plan, and were overseen by the Trial Statistician.

### Ethical review

The trial was reviewed by the North of Scotland Research Ethics Committee (18/NS/0124) and approved on 13 December 2018.

## Results

For this feasibility trial, our main aim was to assess recruitment and retention rates, and to gather qualitative feedback on recruitment and participation.

### Recruitment and retention

Thirty participants were recruited to this trial, fourteen were randomised to Parents InC and sixteen to IY. Recruitment and retention data are presented in Fig 1.

### Sample characteristics

Referring to Table 1, the socioeconomic spread of the sample was skewed towards those living in areas of higher deprivation and 41% of parents exhibited ADHD symptoms (i.e., endorsed ≥4 items on the opening screener questions) on the ASRS screening tool. There was little difference evident between groups at baseline.

Referring to Table 2 below, children of the parents in the sample were predominantly male with a mean age of 9 years. Just under half (44%) had another neurodevelopmental diagnosis at the time of the study.

### Openness to randomisation

No potential participants cited randomisation as a reason for non-participation during recruitment phone-calls with the research team (see Fig 1). Findings from qualitative interviews suggest that 18 of 24 participants interviewed were generally accepting of the principle of randomisation, and were motivated by a belief that attendance at any parenting programme would be of benefit to them. The fact that five participants wanted to attend the group to which they were not randomised (i.e., IY participants attending Parents InC groups) suggests that unwillingness to randomisation was present though not expressed (see Table 3). Interview data recorded the reasons for requested referral to non-randomised group as a) time constraints (a preference for Parents InC as taking less time), and b) ADHD focus of programme (Parents InC was preferred by some due to its focus on ADHD).

### Practitioner referral

Referrer interview data indicated a degree of reluctance to refer based on concerns that referral to the research trial may impact therapeutic relations and perceived irrelevance to certain patients. Others cited workload (falling off their agenda, forgetting to refer) and uncertainty over professional remit as barriers to referral. Of the 52 families assessed for eligibility, only seven (13%) were referred directly by paediatricians/psychiatrists following assessment, and 45 (87%) were already referred to Parents InC and were on a waiting list, and responded to invitation letters from the clinical team.

### Feasibility & acceptability of research procedures and measures

Table 4 demonstrates the mean number of contacts (telephone, text message and letter) per participant from the point of expression of interest to follow up data collection. The quantitative follow-up assessments took the highest number of contacts per participant (mean = 7.4) and, despite researcher concerns that this may be intrusive for participants, overall interview data suggested participants were happy with this level of contact.

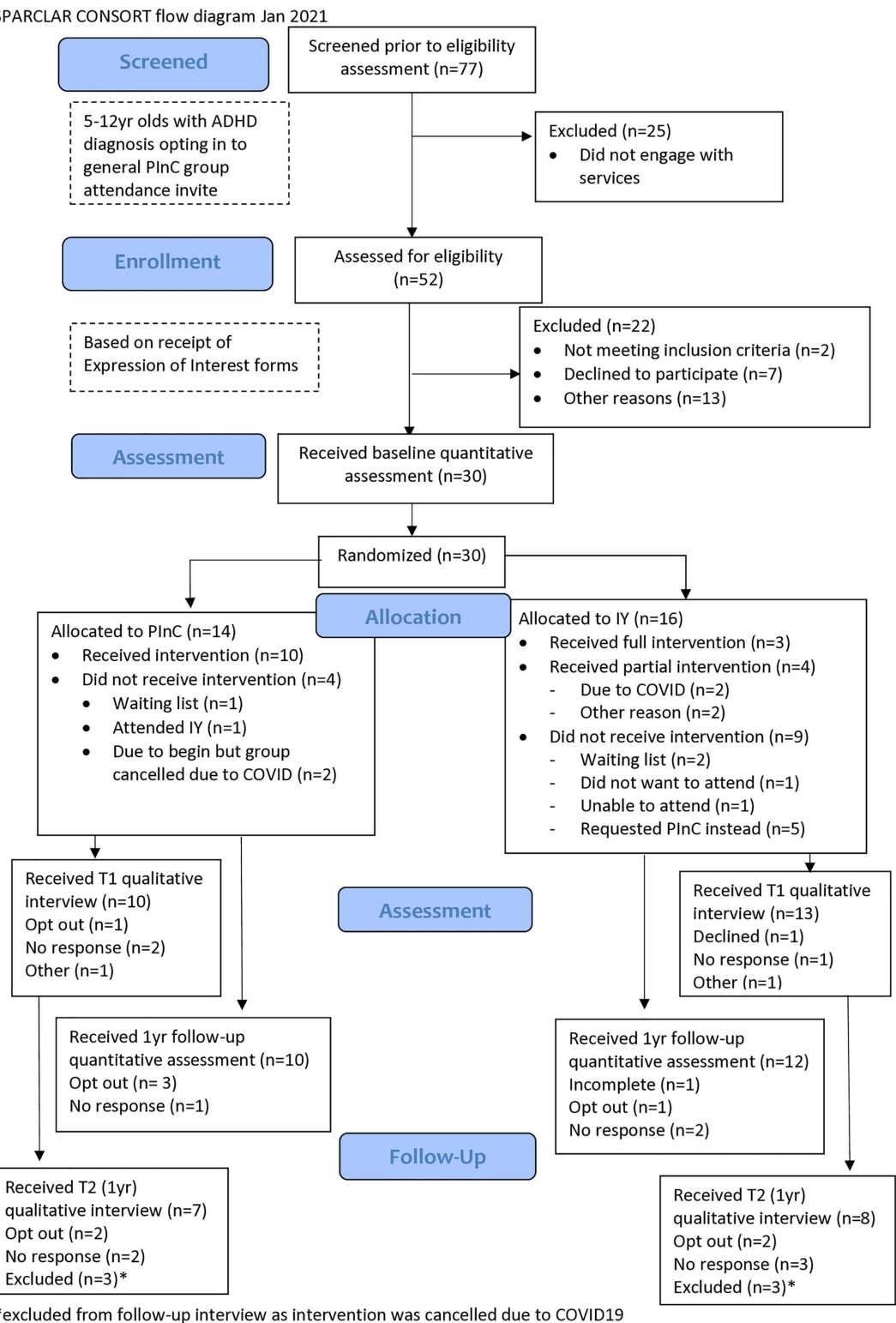

SPARCLAR CONSORT flow diagram Jan 2021

**Screened** — Screened prior to eligibility assessment (n=77)

5-12yr olds with ADHD diagnosis opting in to general PInC group attendance invite

Excluded (n=25)
- Did not engage with services

**Enrollment** — Assessed for eligibility (n=52)

Based on receipt of Expression of Interest forms

Excluded (n=22)
- Not meeting inclusion criteria (n=2)
- Declined to participate (n=7)
- Other reasons (n=13)

**Assessment** — Received baseline quantitative assessment (n=30)

Randomized (n=30)

**Allocation**

Allocated to PInC (n=14)
- Received intervention (n=10)
- Did not receive intervention (n=4)
  - Waiting list (n=1)
  - Attended IY (n=1)
  - Due to begin but group cancelled due to COVID (n=2)

Allocated to IY (n=16)
- Received full intervention (n=3)
- Received partial intervention (n=4)
  - Due to COVID (n=2)
  - Other reason (n=2)
- Did not receive intervention (n=9)
  - Waiting list (n=2)
  - Did not want to attend (n=1)
  - Unable to attend (n=1)
  - Requested PInC instead (n=5)

**Assessment**

Received T1 qualitative interview (n=10)
Opt out (n=1)
No response (n=2)
Other (n=1)

Received T1 qualitative interview (n=13)
Declined (n=1)
No response (n=1)
Other (n=1)

Received 1yr follow-up quantitative assessment (n=10)
Opt out (n= 3)
No response (n=1)

Received 1yr follow-up quantitative assessment (n=12)
Incomplete (n=1)
Opt out (n=1)
No response (n=2)

**Follow-Up**

Received T2 (1yr) qualitative interview (n=7)
Opt out (n=2)
No response (n=2)
Excluded (n=3)*

Received T2 (1yr) qualitative interview (n=8)
Opt out (n=2)
No response (n=3)
Excluded (n=3)*

*excluded from follow-up interview as intervention was cancelled due to COVID19

**Fig 1. CONSORT flow diagram.**

**Table 1. Parent characteristics.**

| | | All (30) | | STUDY ARM PInC (14) | | IY (16) | |
|---|---|---|---|---|---|---|---|
| AGE (YEARS) MEAN (SD) | | 391 | (7.2) | 388 | (7.0) | 393 | (7.5) |
| GENDER N (%) | Male | 1 | (3%) | 1 | (7%) | 0 | (0%) |
| | Female | 29 | (97%) | 13 | (93%) | 16 | (100%) |
| SOCIOECONOMIC DEPRIVATION (SIMD*) N (%) | Q1 (most deprived) | 7 | (23%) | 3 | (21%) | 4 | (25%) |
| | Q2 | 12 | (40%) | 7 | (50%) | 5 | (31%) |
| | Q3 | 3 | (10%) | 1 | (7%) | 2 | (12%) |
| | Q4 | 4 | (13%) | 3 | (21%) | 1 | (6%) |
| | Q5 (least deprived) | 4 | (13%) | 0 | (0%) | 4 | (25%) |
| PARENT ADHD SYMPTOMS (ASRS PART A) N (%) | Yes | 12 | (41%) | 7 | (50%) | 5 | (33%) |
| | No | 17 | (59%) | 7 | (50%) | 10 | (67%) |

* Scottish Index of Multiple Deprivation 2020(SIMD) [53]

**Table 2. Child characteristics.**

| | | Randomised Intervention All (30) | | PInC (14) | | IY (16) | |
|---|---|---|---|---|---|---|---|
| Age(years); Mean (SD) | | 9.1 | (1.5) | 8.9 | (1.8) | 9.3 | (1.2) |
| Gender; N(%) | Male | 26 | (87%) | 10 | (71%) | 16 | (100) |
| | Female | 4 | (13%) | 4 | (29%) | 0 | (0) |
| Time since diagnosis (years) | | 4.6 | (3.4) | 3.6 | (3.4) | 5.7 | (3.3) |
| On ADHD medication | No | 12 | (40%) | 6 | (43%) | 6 | (37.5%) |
| | Yes | 18 | (60%) | 8 | (57%) | 10 | (62.5%) |
| Time since starting medication (months) | | 9.0 | (11.7) | 16.1 | (14.6) | 3.4 | (3.7) |
| Any other neurodevelopmental diagnosis | No | 15 | (56%) | 6 | (46%) | 9 | (64%) |
| | Yes | 12 | (44%) | 7 | (54%) | 5 | (36%) |

**Table 3. Randomisation and group attendance summary.**

| | N | Proportion |
|---|---|---|
| Participants consented & randomised | 30 | |
| Participants attended randomisation group (out of participants randomised) | 17 | 57% |
| Participants requested non-randomisation group (out of participants randomised) (n = 5 randomised to IY, n = 0 randomised to Parents InC) | 5 | 13% |
| Participants attended both groups (out of subjects randomised) | 1 | 3% |
| Participants did not attend any group (out of subjects randomised) | 7 | 23% |

> *"Yes, so the phone call and the text messages are fine, but I do appreciate that sometimes I am a bit hard to get a hold of because of work and stuff, but it has been fine, yeah… Aye, definitely; the reminder texts are great!" (female parent 1, follow-up interview).*

Participant motivation to engage with the research was explored during interviews, the key factor being the length of the child's journey to diagnosis; parents reported that having waited so long for a diagnosis, they were willing to participate in any activity which would support their child's ongoing development.

**Table 4. Mean number of contacts per participant.**

| | No. of contacts | |
| --- | --- | --- |
| | Total (n participants) | Mean (SD) per participant |
| Quantitative baseline | 99 (30) | 3.3 (1.9) |
| Qualitative baseline | 139 (29) | 4.8 (3.9) |
| Quantitative Follow up | 221 (30) | 7.4 (6.5) |
| Qualitative Follow up | 90 (21) | 4.3 (3.2) |
| Total | 549 | 18.3 (9.7) |

The acceptability of research procedures centred around three key themes during interview:

1) researcher approach: Participants commented on the importance of researchers' empathic interactions and flexibility of data collection methods (particularly relating to time of day/number of sessions and location of data collection). Both phone and face-to-face interviews were reported to be acceptable to qualitative interview participants.

"…it was fine, and you came to the house last time, that was absolutely fantastic as I didn't have to go anywhere, it worked round me, it was relaxed, same as this time, it has just been absolutely fine." (female parent 2, follow-up interview).

2) study information: The majority of participants felt the study was well explained and facilitated by the research team.

"(…did it go to plan for you? Was it what you expected?) Yeah, it was and more. I think we had lots of support overall, any questions could get answered at any time and we've been given enough information…" (female parent 3, follow-up interview).

3) timing with intervention: Due to waiting times for randomised programmes one parent opted out of attending a programme, and two others attended programmes they weren't randomised to while waiting.

"I was quite happy to go on them and I was looking forward to it, but then it took too long as in it took too long to start with, to come round, so I think we must have waited may be almost ten months before we actually got put on a course and then obviously it was then for quite a long period of time, I think it was 14 weeks, and it was over lunch time, and it was for may be about two hours, and that didn't really suit us…" (female parent 4, follow-up interview).

Participants' views of the research measures were generally positive, and many found the questions familiar from clinical diagnostic interviews. Some participants reported feeling emotional or upset by discussing family medical history and detailing the everyday challenges they face in parenting their child with ADHD, though this did not cause sufficient distress to prevent their ongoing participation.

**Parenting Sense of Competence Scale (PSOC) results at follow-up**

Data from the Parenting Sense of Competence scale at baseline and follow-up for each intervention arm is reported in Table 5. Regression analysis suggests that participants who were randomised to the Parents InC group reported a higher sense of parenting competency following intervention than those randomised to the IY group.

**Table 5. PSOC data from participants with baseline and follow-up data only.**

| | | All | | Randomised intervention | | | |
|---|---|---|---|---|---|---|---|
| | | | | PinC | | IY | |
| | $N_{OBS}$ | 18 | | 9 | | 9 | |
| Baseline | Mean (SD) | 62.5 | (7.0) | 64.2 | (8.0) | 60.7 | (5.8) |
| Follow-up | Mean (SD) | 65.5 | (7.1) | 68.9 | (7.1) | 62.1 | (5.7) |
| Change from baseline | Mean (SD) | 3.0 | (6.9) | 4.7 | (6.6) | 1.4 | (7.3) |

*Linear regression analysis predicting change from baseline from randomised intervention group adjusting for baseline PSOC, child age and gender*

| | Adjusted mean difference (95% CI) | | | | | p-value | |
|---|---|---|---|---|---|---|---|
| **IY vs. PinC** | −7.44 (−14.14, −0.74) | | | | | 0.049 | |

## Statistical power for a future definitive RCT

Recruitment and retention was reasonably good in this study, but 43% of participants did not go on to receive the intervention they were randomised to. Although we would seek to mitigate the possibility of participants opting for the comparison intervention in a definitive trial (see limitations), we have chosen to retain a conservative approach to sample size calculation. We estimate that it would be feasible to detect 0.5 of a SD change in PSOC scores at 90% power with 191 participants in each arm. Based on our recruitment and follow-up rates, we would expect to have to recruit around 500 participants to reach this figure. A future trial could potentially stratify using variables such as parental ADHD status (ASRS cut-off) and SIMD.

## All psychometric results at baseline and follow-up

A summary of all psychometric mean scores is presented in supplementary S1 Table.

Overall, the parent focussed outcomes (PSOC and parental GHQ) suggest the largest effects between intervention arms. Child-focussed outcome measures and quality of life suggested little difference between groups.

## Economic summary

Costs per group were calculated at £14,957 for Parents InC; £26,489 for IY. In terms of associated health service costs: IY participants had more outpatient visits in 12 months post-randomisation, but use of other services was similar across arms.

## Discussion

The findings of this feasibility study support the intention of exploring the research questions in a future substantive randomised controlled trial (RCT). The trial has demonstrated reasonable recruitment and excellent retention rates. Participant feedback on research procedures and methods was positive, and qualitative interviews provided a rich source of learning points to take forward into a future definitive trial. A secondary focus of this trial has been on parent outcomes following group intervention; this focus is upheld by results suggesting the largest between-group effect sizes on the parental sense of competence scale and parental General Health Questionnaire.

Our study supports those of previous work which highlights the need for careful planning and consideration of pragmatic barriers as part of the research design. As both intervention groups in the present study were part of usual services, recruitment to the research trial was dependent on referrals from NHS staff/ access to the Parents InC waiting list. Interview data demonstrated a level of gate-keeping from referring practitioners based on misconceptions about the trial and heavy workloads resulting in de-prioritising of referrals. In a definitive RCT, the trial research team would retain full

responsibility for recruitment by negotiating access to routine data to define the target population, as has been successful in other RCTs we have conducted [41]. Whilst evaluating ongoing health services can be financially beneficial, there are doubtless compromises made to the integrity of the research project.

A positive aspect to recruitment was the interview method used in the informed consent and baseline measures process. Although this may be an intensive method, it may be a worthwhile investment in the context of this participant group and service setting.

It is interesting to note that the theme of gate-keeping within health care systems and systemic barriers to engagement in research and health services is often acknowledged [54]. The use of the term "hard to reach" has long been misused and conveniently places responsibility upon those families to whom the term applies; leaving the systems and institutions who coined the term on a moral high ground, having "done their bit" and produced a beautifully crafted yet inaccessible programme. The developer of the Incredible Years programme shared the following insight:

*"Such families have been described as unmotivated, resistant, unreliable, disengaged, chaotic...unlikely candidates for this kind of treatment — in short, unreachable. However, these families might well describe traditional clinic-based programs as 'unreachable'. Clinical programs may be too far away from home, too expensive, insensitive, distant, inflexible in terms of scheduling or content, foreign in terms of language...blaming or critical of their lifestyle. Perhaps this population has been 'unreachable' not because of their own characteristics, but because of the characteristics of the interventions they have been offered."* [55]

Our qualitative work has allowed a deeper understanding of barriers and facilitators for this population, and any future trial could benefit from incorporating this sort of exploration into pre-protocol work. For example, Chacko et al [56] showed that parental perception of the relevance of behavioural parent training was of particular importance in their sample, and that parental depression and stress levels were not significantly associated with engagement. Similarly, Bazier et al [57] demonstrated the importance of parental perceptions in help seeking once pragmatic barriers had been overcome. There is scope for more nuanced exploration within the field; recent advances such as mindfulness-based parenting interventions [58] may be popular but demonstrate mixed efficacy in parental outcomes and, whilst behavioural parenting interventions are broadly well supported by the evidence, the particular components of these programmes which drive efficacy are now being brought under scrutiny [24].

While recruitment to this trial was lower than anticipated, retention was far higher. Once participants were successfully engaged, they expressed willingness to participate and motivation to engage with both the quantitative and qualitative components of the trial, therefore we anticipate higher recruitment rates if access to the entire target population was made possible through access to routine data. An interesting feature in participant motivation to engage was the length of the child's journey to diagnosis; parents reported that having waited so long for a diagnosis, they were willing to participate in any activity which would support their child's ongoing development. This highlights the importance of understanding not only population characteristics, but also the context within which a research trial is to be conducted. Whether system variables such as length of waiting times for service support has an impact on outcomes of any parenting intervention should be assessed.

## Limitations

As discussed previously, the sample size for this feasibility study was small due to reliance on practitioner referral and staffing issues. Parents InC is the first-line parenting programme for parents of a child with ADHD in NHS Fife. This intervention was therefore available to all participants, even if randomised to Incredible Years, allowing participants to request re-referral to Parents InC during the trial and this might have biased our findings regarding participant preferences. Not only did this result in a low sample size, only 57% (17/30) of those randomised received the intervention they

were randomised to. This may be unlikely in a different trial context, but does impact on the application of these findings to the planning of a definitive trial. A future trial would be best placed within a setting where no or very limited current service provision exists, so that there is no ethical requirement to allow participants to select either intervention, regardless of their study allocation. The characteristics of the interventions offered (Incredible Years lasting 14 sessions and Parents InC lasting six sessions) may influence systemic investment in parenting interventions (at both health-board and family-level), a factor worth exploring more comprehensively in a future trial. Finally, the need to only include those with sufficient English language proficiency limits the generalisability of these findings: a future trial should ensure sufficient funding to allow other languages to be used and consider how best to address potential cultural differences in response to measures and intervention content within the target population.

## Conclusions

In a public healthcare system, it is imperative that treatments and interventions are both efficacious and cost-effective. The current research indicates that parental sense of competence and general health may be enhanced by attendance at a Parents InC group, and further research is required to explore this effect on a larger scale. The present findings show that recruitment and retention to a large definitive trial of Parents InC vs a care as usual intervention such as Incredible Years is feasible, but that certain conditions need to be carefully considered to ensure the research is accessible. Despite personal and situational barriers, populations such as reported here are not "hard" to reach and are often highly motivated to engage in services and research – it is our role as clinicians and researchers to commit to removing those barriers which inhibit their engagement.

## Supporting information

**S1 Table. Between-group difference secondary outcome measures.**
(DOCX)

**S2 File. Statistical Analysis Plan.**
(DOCX)

## Acknowledgments

We are extremely grateful to the parents who gave their time to take part in research assessments and interviews. Our Trial Steering Committee, including Geraldine Mynors, Dr Christine Puckering, and Linda Frazer (Patient and Public Involvement representative) offered valuable discussion and reflection. Thanks to Fatene Ismail, Charlotte Paterson, and Paula Regener who previously worked on the trial. We appreciate the input of Martina Messow for writing the original Statistical analysis plan and supervising descriptive and outcome analyses, and of Ting Ting Wu for conducting health economic analysis. Our co-investigators helped shaped the design of the study: Nick Watson, Rachael Wood. Finally, thanks to Dr Hilary Maddox, Consultant Clinical Psychologist and Head of Child and Family Psychology Service, NHS Fife, and Gordon Brown, Clinical Lead and CEO of ADHD Direct, who developed the Parents InC programme and provided information on the theory of change. Dr Maddox also hosted the study within her service.

## Author contributions

**Conceptualization:** Emma McIntosh, Caroline Haig, Christopher Gillberg, Helen Minnis, Lucy Thompson.

**Data curation:** Fiona Sim, Lindsay Dalgarno, Emma McIntosh, Caroline Haig.

**Formal analysis:** Fiona Sim, Lindsay Dalgarno, Emma McIntosh, Caroline Haig, Patrycja Duklas, Alex McConnachie.

**Funding acquisition:** Emma McIntosh, Caroline Haig, Christopher Gillberg, Helen Minnis, Lucy Thompson.

**Investigation:** Fiona Sim, Lindsay Dalgarno, Emma McIntosh, Caroline Haig, Alex McConnachie, Christopher Gillberg, Helen Minnis, Lucy Thompson.

**Methodology:** Lindsay Dalgarno, Emma McIntosh, Caroline Haig, Alex McConnachie, Christopher Gillberg, Helen Minnis, Lucy Thompson.

**Project administration:** Fiona Sim, Lindsay Dalgarno, Lucy Thompson.

**Supervision:** Fiona Sim, Emma McIntosh, Caroline Haig, Alex McConnachie, Christopher Gillberg, Helen Minnis, Lucy Thompson.

**Validation:** Fiona Sim, Lindsay Dalgarno, Emma McIntosh, Caroline Haig, Patrycja Duklas, Lucy Thompson.

**Visualization:** Fiona Sim.

**Writing – original draft:** Fiona Sim, Lindsay Dalgarno, Patrycja Duklas, Lucy Thompson.

**Writing – review & editing:** Fiona Sim, Lindsay Dalgarno, Emma McIntosh, Caroline Haig, Patrycja Duklas, Alex McConnachie, Christopher Gillberg, Helen Minnis, Lucy Thompson.

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
