## [Decision Letter · Decision Letter 0]

18 Nov 2024

Dear Dr. Thompson,

Thank you for submitting your manuscript to PLOS ONE. After careful consideration, we feel that it has merit but does not fully meet PLOS ONE’s publication criteria as it currently stands. Therefore, we invite you to submit a revised version of the manuscript that addresses the points raised during the review process.

We look forward to receiving your revised manuscript.

Kind regards,

Mu-Hong Chen, M.D., Ph.D.

Academic Editor

PLOS ONE

Journal Requirements:

2. In the online submission form, you indicated that your data will be submitted to a repository upon acceptance. We strongly recommend all authors deposit their data before acceptance, as the process can be lengthy and hold up publication timelines. Please note that, though access restrictions are acceptable now, your entire minimal dataset will need to be made freely accessible if your manuscript is accepted for publication. This policy applies to all data except where public deposition would breach compliance with the protocol approved by your research ethics board. If you are unable to adhere to our open data policy, please kindly revise your statement to explain your reasoning and we will seek the editor's input on an exemption.

Reviewers' comments:

Reviewer's Responses to Questions

**Comments to the Author**

1. Is the manuscript technically sound, and do the data support the conclusions?

Reviewer #1: Partly

Reviewer #2: No

2. Has the statistical analysis been performed appropriately and rigorously?

Reviewer #1: Yes

Reviewer #2: Yes

3. Have the authors made all data underlying the findings in their manuscript fully available?

Reviewer #1: Yes

Reviewer #2: Yes

4. Is the manuscript presented in an intelligible fashion and written in standard English?

Reviewer #1: Yes

Reviewer #2: Yes

Reviewer #1: Summary of Manuscript:

The manuscript under review presents findings from a feasibility trial comparing different parenting programs aimed at parents of children diagnosed with ADHD. The study challenges the notion that families with neurodevelopmental diagnoses are inherently 'hard to reach' by evaluating the accessibility and effectiveness of various parenting interventions.

Overall Evaluation:

The manuscript provides valuable insights into the accessibility and effectiveness of parenting programs for families with children with ADHD. The findings have the potential to influence future research and practice in this field. However, several major revisions are necessary to improve the clarity, robustness, and impact of the research.

Major Revisions Required:

1. Clarification of Objectives:

o Issue: The manuscript lacks a clear and detailed explanation of the study objectives.

o Recommendation: Revise the introduction to explicitly state the research questions. Clearly articulate what the study aims to demonstrate regarding the reach and effectiveness of different parenting programs.

2. Methodology Details:

o Issue: There is insufficient detail about the study design, participant selection, and intervention procedures.

o Recommendation: Provide a more comprehensive description of the study design, including randomization procedures, inclusion and exclusion criteria, and detailed descriptions of the parenting programs compared. This will help in assessing the robustness of the methodology.

3. Statistical Analysis:

o Issue: The statistical methods used are not adequately described, and there is a lack of justification for the chosen analyses.

o Recommendation: Elaborate on the statistical techniques employed, including how they were selected and why they are appropriate for the data. Provide justification for any assumptions made and discuss how missing data were handled.

4. Results Presentation:

o Issue: The presentation of results is somewhat disorganized and lacks coherence.

o Recommendation: Reorganize the results section to clearly present the findings for each parenting program in a systematic manner. Use tables and figures to summarize key results and make comparisons more accessible.

5. Discussion and Interpretation:

o Issue: The discussion does not sufficiently interpret the findings in the context of existing literature and does not address potential limitations.

o Recommendation: Expand the discussion to contextualize the findings within the broader literature on ADHD and parenting programs. Address the study’s limitations in detail and suggest how these limitations might affect the interpretation of the results.

6. Implications for Practice:

o Issue: The implications for practice and policy are not clearly articulated.

o Recommendation: Articulate the practical implications of the findings for clinicians and policymakers. Provide specific recommendations based on the results, and discuss how these recommendations might influence future interventions for families with neurodevelopmental diagnoses.

7. References and Literature Review:

o Issue: The literature review is outdated and does not incorporate recent studies.

o Recommendation: Update the literature review to include recent research relevant to ADHD parenting programs. Ensure that all cited references are current and pertinent to the study.

8. Formatting and Presentation:

o Issue: The manuscript requires formatting adjustments to meet journal guidelines.

o Recommendation: Revise the manuscript to adhere to the journal’s formatting requirements, including section headings, font size, and citation style.

Conclusion:

The manuscript presents important findings that could contribute significantly to the field of parenting programs for families with children with ADHD. However, substantial revisions are required to enhance the clarity, methodological rigor, and overall impact of the study. Addressing the points outlined above will strengthen the manuscript and better position it for publication.

Recommendation:

Revise and resubmit for further review.

Additional Comments:

• Ensure that all ethical considerations are clearly addressed.

Consider adding a summary of the study’s contributions to the field at the end of the discussion.

Reviewer #2: The idea of comparing the Parents InC and Incredible Years (IY) programs in this trial is interesting, as it explores two parenting interventions aimed at different aspects of managing ADHD in children. The design and intention to assess the feasibility of a future, larger-scale RCT is commendable and well-structured. However, there are several limitations that could impact the reliability and interpretation of the findings, which should be addressed to improve the robustness and transparency of the results.

One key concern is the lack of detailed information about the children's baseline psychiatric conditions in each group, particularly the severity of ADHD and any comorbidities such as anxiety, depression, or autism spectrum disorder. Understanding the severity of the children's symptoms and the presence of any other conditions is essential because these factors can significantly influence treatment outcomes. For instance, if one group has children with more severe ADHD or additional comorbid conditions, it could confound the results and make it difficult to draw accurate conclusions about the relative effectiveness of the two parenting programs. Furthermore, the different target audiences of the two programs also pose a challenge when interpreting the results. As noted in the manuscript, IY addresses not only ADHD, but also oppositional defiant disorder (ODD), conduct disorder (CD), and anxiety. Meanwhile, Parents InC is more focused on ADHD specifically. Given that these conditions can significantly affect parenting dynamics and the child-parent relationship, comorbidities could be an important factor in explaining the differences in outcomes between the two groups. The study would benefit from a deeper exploration of these potential confounders, especially as IY may have broader applications and address a wider range of behavioral and emotional challenges.

Additionally, as the study only reports on parent-focused outcomes like parental sense of competence (PSOC) and the General Health Questionnaire (GHQ), the children's treatment — particularly medications — is another important factor that is notably missing from the study. Pharmacological treatment is the first-line ADHD management for children over the age of six, and since both ADHD medication and behavioral interventions can independently influence outcomes, it would have been crucial to know the treatment regimens for the children in both groups. Without this information, it’s difficult to ascertain whether any observed effects are due to the parenting interventions alone or influenced by pharmacological treatments that were not accounted for.

**Do you want your identity to be public for this peer review?** For information about this choice, including consent withdrawal, please see our Privacy Policy

Reviewer #1: No

Reviewer #2: No

---

## [Author Response · Author response to Decision Letter 1]

16 Jan 2025

We have addressed reviewer comments in the uploaded file 'response to reviewers'.

---

## [Decision Letter · Decision Letter 1]

23 Feb 2025

Dear Dr. Thompson,

Thank you for submitting your manuscript to PLOS ONE. After careful consideration, we feel that it has merit but does not fully meet PLOS ONE’s publication criteria as it currently stands. Therefore, we invite you to submit a revised version of the manuscript that addresses the points raised during the review process.

We look forward to receiving your revised manuscript.

Kind regards,

Mu-Hong Chen, M.D., Ph.D.

Academic Editor

PLOS ONE

Reviewers' comments:

Reviewer's Responses to Questions

**Comments to the Author**

Reviewer #1: (No Response)

Reviewer #2: All comments have been addressed

2. Is the manuscript technically sound, and do the data support the conclusions?

Reviewer #1: Partly

Reviewer #2: Yes

3. Has the statistical analysis been performed appropriately and rigorously?

Reviewer #1: Yes

Reviewer #2: Yes

4. Have the authors made all data underlying the findings in their manuscript fully available?

Reviewer #1: Yes

Reviewer #2: Yes

5. Is the manuscript presented in an intelligible fashion and written in standard English?

Reviewer #1: No

Reviewer #2: Yes

Reviewer #1: General Overview

The manuscript is well-structured, with clear sections and a comprehensive description of methods, results, and implications. The topic is relevant for both researchers and practitioners addressing ADHD interventions.

Areas for Improvement and Comments

The introduction could benefit from a clearer articulation of the specific research gap addressed by the study. While the importance of ADHD parenting programs is emphasized, the distinct advantages of comparing Parents InC and IY are less explicit.

Consider discussing broader implications for other neurodevelopmental conditions given the overlap in symptomatology such as Autism Spectrum Disorder.

The rationale for excluding parents with low English proficiency could be elaborated further.

While the randomization method is described, it would be helpful to clarify whether participants were blinded to their allocation and how this might have influenced outcomes.

The choice of the Parenting Sense of Competence Scale (PSOC) as the primary measure is appropriate but could be supplemented with observational measures of parent-child interaction for future trials.

The sample size of 30 participants is acknowledged as a limitation. While this is typical for a feasibility study, discussing strategies to improve recruitment such as more robust outreach efforts or incentives) would strengthen this section.

Table 3 highlights participants requesting non-randomized groups. This preference deserves more exploration in the discussion, as it reflects on the acceptability of the interventions.

The discussion of barriers to recruitment and retention is insightful but could benefit from greater emphasis on potential systemic solutions such as improving practitioner referral pathways or addressing gatekeeping.

The authors critique the term "hard to reach" and advocate for a systems-level perspective. This is a compelling argument, but more concrete recommendations for service redesign would add practical value.

The differences in session length and focus between Parents InC (6 sessions) and IY (14 sessions) should be more critically evaluated. Are these differences in program structure likely to impact scalability and accessibility?

While the authors acknowledge the limited generalizability due to language restrictions and reliance on existing services, further discussion on how these limitations can be mitigated in future research would be valuable.

The CONSORT flow diagram is helpful but could be improved by visually emphasizing the drop-out points and reasons.

Tables summarizing qualitative themes could be added to complement the narrative findings.

Ensure consistent use of terminology such as "Parents InC" vs. "PInC".

Some references are slightly dated such as DSM-5 reference from 2022. Consider incorporating more recent studies to highlight ongoing developments in ADHD research and interventions.

Reviewer #2: Introduction

The rationale for comparing Parents InC to Incredible Years is well articulated; however, further explanation is needed regarding why these particular interventions were chosen. Was Parents InC already widely used? Why was it not compared to another ADHD-specific program?

Results

The recruitment and retention data are well presented; however, the study does not sufficiently explore the reasons behind the high dropout rate or participation in non-randomized groups.

Conclusion

It would be helpful to also address how to use data obtained from this study to guide the future ECT to explore factors relevant to "hard to reach" and "gate-keeping" phenomena.

**Do you want your identity to be public for this peer review?** For information about this choice, including consent withdrawal, please see our Privacy Policy

Reviewer #1: No

Reviewer #2: No

---

## [Author Response · Author response to Decision Letter 2]

19 Mar 2025

Please see uploaded response to reviewers document

---

## [Decision Letter · Decision Letter 2]

17 Apr 2025

Families with neurodevelopmental diagnoses are not ‘Hard to Reach’: Findings from a feasibility trial comparing parenting programmes for parents of children with ADHD

PONE-D-24-36397R2

Dear Dr. Lucy Thompson,

We’re pleased to inform you that your manuscript has been judged scientifically suitable for publication and will be formally accepted for publication once it meets all outstanding technical requirements.

Kind regards,

Mu-Hong Chen, M.D., Ph.D.

Academic Editor

PLOS ONE

Additional Editor Comments (optional):

Reviewers' comments:

Reviewer's Responses to Questions

**Comments to the Author**

Reviewer #2: All comments have been addressed

2. Is the manuscript technically sound, and do the data support the conclusions?

Reviewer #2: Yes

3. Has the statistical analysis been performed appropriately and rigorously?

Reviewer #2: Yes

4. Have the authors made all data underlying the findings in their manuscript fully available?

Reviewer #2: Yes

5. Is the manuscript presented in an intelligible fashion and written in standard English?

Reviewer #2: Yes

Reviewer #2: (No Response)

**Do you want your identity to be public for this peer review?** For information about this choice, including consent withdrawal, please see our Privacy Policy

Reviewer #2: No

---

## [Editor Report · Acceptance letter]

PONE-D-24-36397R2

PLOS ONE

Dear Dr. Minnis,

I'm pleased to inform you that your manuscript has been deemed suitable for publication in PLOS ONE. Congratulations! Your manuscript is now being handed over to our production team.

Kind regards,

on behalf of

Dr. Mu-Hong Chen

Academic Editor

PLOS ONE